# Extent, impact, and mitigation of batch effects in tumor biomarker studies using tissue microarrays

Konrad H Stopsack[1,2]*, Svitlana Tyekucheva[3,4], Molin Wang[1,3,5], Travis A Gerke[6], J Bailey Vaselkiv[1], Kathryn L Penney[1,5], Philip W Kantoff[2], Stephen P Finn[7,8], Michelangelo Fiorentino[1,9], Massimo Loda[10], Tamara L Lotan[11], Giovanni Parmigiani[3], Lorelei A Mucci[1]

[1]Department of Epidemiology, Harvard T.H. Chan School of Public Health, Boston, United States; [2]Department of Medicine, Memorial Sloan Kettering Cancer Center, New York, United States; [3]Department of Biostatistics, Harvard T.H. Chan School of Public Health, Boston, United States; [4]Department of Data Science, Dana-Farber Cancer Institute, Boston, United States; [5]Channing Division of Network Medicine, Department of Medicine, Brigham and Women's Hospital and Harvard Medical School, Boston, United States; [6]Department of Cancer Epidemiology, Moffitt Cancer Center, Tampa, United States; [7]Department of Pathology, St. James's Hospital, Dublin, Ireland; [8]Trinity College, Dublin, Ireland; [9]Pathology Unit, Addarii Institute, S. Orsola-Malpighi Hospital, Bologna, Italy; [10]Department of Pathology, Weill Cornell Medical College, New York, United States; [11]Department of Pathology, Johns Hopkins Medical Institutions, Baltimore, United States

**Abstract** Tissue microarrays (TMAs) have been used in thousands of cancer biomarker studies. To what extent batch effects, measurement error in biomarker levels between slides, affects TMA-based studies has not been assessed systematically. We evaluated 20 protein biomarkers on 14 TMAs with prospectively collected tumor tissue from 1448 primary prostate cancers. In half of the biomarkers, more than 10% of biomarker variance was attributable to between-TMA differences (range, 1–48%). We implemented different methods to mitigate batch effects (R package *batchtma*), tested in plasmode simulation. Biomarker levels were more similar between mitigation approaches compared to uncorrected values. For some biomarkers, associations with clinical features changed substantially after addressing batch effects. Batch effects and resulting bias are not an error of an individual study but an inherent feature of TMA-based protein biomarker studies. They always need to be considered during study design and addressed analytically in studies using more than one TMA.

*For correspondence:
stopsack@post.harvard.edu

## Editor's evaluation

Tissue microarrays (TMA) have become a mainstay in clinical and basic research, for both discovery and validation of biomarkers. This manuscript provides relevant methodological considerations for cancer researchers investigating tissue-biomarkers using TMAs. A comprehensive investigation was conducted using a combination of analytic approaches using empirical data and simulated data to support key findings and conclusions. The authors approach the possible sampling variation in a thoughtful way, not only quantifying the issue systematically, but working towards a solution.

## Introduction

Tissue microarrays (TMAs) were first developed in the 1990s as an efficient way to examine tissue-based biomarkers (*Kononen et al., 1998*). Since then, TMAs have been used in thousands of studies to evaluate histologic and molecular biomarkers, mostly in cancer tissue. Individual TMAs consist of cylindrical cores from hundreds of tissue samples embedded in one recipient block (*Kononen et al., 1998*; *Kallioniemi et al., 2001*). Studies often include more than one TMA. Even when biomarker assays are well standardized and run conditions are diligently kept fixed, some TMA slides (batches) may have measurements systematically too low or too high, and some batches may have wider spread around the true values of the biomarker than others. In general, such batch effects can have a profound impact on the validity of biomarker studies, such as those using RNA microarrays (*Tworoger and Hankinson, 2006*; *Leek et al., 2010*). Contrary to popular belief, whether such measurement error induces upward or downward bias in results is not guaranteed to follow simple heuristics (*van Smeden et al., 2020*).

Whether and to what extent TMAs are affected by batch effects has not been empirically assessed. TMAs pose unique challenges. For example, when tumor tissue is collected prospectively for inclusion on TMAs, tumor characteristics may differ between batches due to nonrandom assignment of cases, as well as temporal trends in tumor risk factors, screening, and diagnosis. Differences in tissue processing or storage across tissue specimens may have differential impact on biomarkers. Including calibration samples for quality control is also more challenging for TMAs than, for example, assaying of blood samples, because repeat sections from a tumor may differ due to intratumoral heterogeneity rather than only batch effects.

In this study, we assess batch effects in a large set of centrally constructed TMAs from prostate cancer tissue from 1448 men in two nationwide cohort studies. We quantify the extent to which protein biomarker variation could be explained by batch effects. We probe different methods for mitigating batch effects while maintaining true, "biological," between-TMA variation, including in a plasmode simulation. Finally, we demonstrate the impact of handling batch effects on commonly performed biomarker analyses.

## Results

### Extent and type of batch effects

To evaluate the presence of batch effects in studies using TMAs, we studied tumor tissue from 1448 men with primary prostate cancer on 14 TMAs (labeled "A" through "N"), each including multiple tumor cores from 47 to 158 patients per TMA (*Figure 1*). Multiple cores from the same tumor (usually 3) were always located on the same TMA.

TMAs were used to quantify 20 protein biomarkers (*Figure 2*). Biomarker values showed noticeable between-TMA variation, despite immunohistochemical staining having been conducted at the same time for all 14 TMAs. We estimated that across the 20 biomarkers, between-TMA variation explained between 1% and 48% of the overall variation in biomarker levels (intraclass correlation coefficient, ICC), with half of the biomarkers having ICCs greater than 10% (*Figure 2*).

In an example biomarker, estrogen receptor alpha in nuclei of stromal cells (*Figure 3*), the means of the most extreme TMAs differed by 2.2 standard deviations in intensity of expression and variances differed up to 9.3-fold. Other biomarkers showed similar between-TMA variation by magnitude and by which TMAs had the most extreme values (*Figure 4A*). Likewise, we observed that not only means, but also variances of biomarker levels differed between TMAs, although patterns of heteroskedasticity appeared weaker than for means (*Figure 4—figure supplement 1*). In contrast, we found little evidence for more complex patterns of batch effects, such that tumors with specific grade, stage, or year of diagnosis would have been particularly affected by between-TMA differences (*Supplementary file 1a*). Nevertheless, observations from the same TMAs tended to be clustered together when projected onto the first two principal components, capturing 27% of the variance in all biomarkers (*Figure 4B*).

Some biomarkers were stained using automated staining systems, other stains were done manually (*Figure 2*). Moreover, the method of scoring, including human (eye) scoring and computer-assisted quantification, differed between biomarkers, as did the main quantitative score, typically a measure of staining intensity, a proportion of cells above an intensity threshold, or a combination of both

**eLife digest** To understand cancer, researchers need to know which molecules tumor cells use. These so-called 'biomarkers' tag cancer cells as being different from healthy cells, and can be used to predict how aggressive a tumor may be, or how well it might respond to treatment.

A popular technique for assessing biomarkers across multiple tumors is to use tissue microarrays. This involves taking samples from different tumors and embedding them in a block of wax, which is then cut into micro-thin slices and stained with reagents that can detect specific biomarkers, such as proteins. Each block contains hundreds of samples, which all experience the same conditions. So, any patterns detected in the staining are likely to represent real variations in the biomarkers present.

Many cancer studies, however, often compare samples from multiple tissue microarrays, which may increase the risk of technical artifacts: for example, staining may look stronger in one batch of tissue samples than another, even though the amount of biomarker present in these different arrays is roughly the same. These 'batch effects' could potentially bias the results of the experiment and lead to the identification of misleading patterns.

To evaluate how batch effects impact tissue microarray studies, Stopsack et al. examined 14 wax blocks which contained tumor samples from 1,448 men with prostate cancer. This revealed that for some biomarkers, but not others, there were noticeable differences between tissue microarrays that were clearly the result of batch effects. Stopsack et al. then tested six different ways of fixing these discrepancies using statistical methods. All six approaches were successful, even if the arrays included tumors with different characteristics, such as tumors that had been diagnosed more or less recently.

This work highlights the importance of considering batch effects when using tissue microarrays to study cancer. Stopsack et al. have used their statistical approaches to develop freely available software which can reduce the biases that sometimes arise from these technical artifacts. This could help researchers avoid misleading patterns in their data and make it easier to detect real variations in the biomarkers present between tumor samples.

(*Figure 2*). Notably, between-TMA differences were present with any of these approaches. For example, batch effects were not only present when considering intensities of biomarker staining, as for the estrogen receptor alpha and beta example. Even when setting cutoffs for staining visible by eye and quantifying the number of stain-positive cells, 8% (95% confidence interval [CI], 2–15) of variance in estrogen receptor alpha positivity and 27% (95% CI, 11–42) of estrogen receptor beta positivity were attributable to between-TMA variation (*Figure 2—figure supplement 1*). Our data do not allow distinguishing which of these approaches, if any, were less prone to batch effects.

In summary, we observed a large and concerning degree of between-TMA variation for several biomarkers that were quantified using different approaches, suggesting that addressing batch effects could significantly impact scientific inference.

## Source of batch effects

The noticeable proportion of variance attributable to TMAs could have two possibly co-existing explanations. First, that between-TMA differences in biomarkers reflect different patient and tumor characteristics that need to be retained. Second, that between-TMA differences are artifacts due to systematic measurement error that need to be removed (batch effects).

In support of the first hypothesis, there were noticeable differences in patient and tumor characteristics between TMAs that are likely associated with biomarker levels (*Figure 1*). Along with a 14-year range between the per-TMA medians of cancer diagnosis year, there were differences in the proportion of tumors with a Gleason score of 8 or higher (between 11% and 33%) and rates of lethal disease (between 2 and 16 events per 1000 person-years of follow-up).

In support of the second hypothesis, we observed that certain TMAs had consistently higher or lower biomarker values for the majority of tested biomarkers (*Figure 4A*). For example, the same batches that showed higher-than-average biomarker values for stathmin also had higher-than-average values for PTEN. This example is noteworthy because both markers were assayed together on the same section of each TMA using multiplex immunofluorescence, and stathmin would be expected to

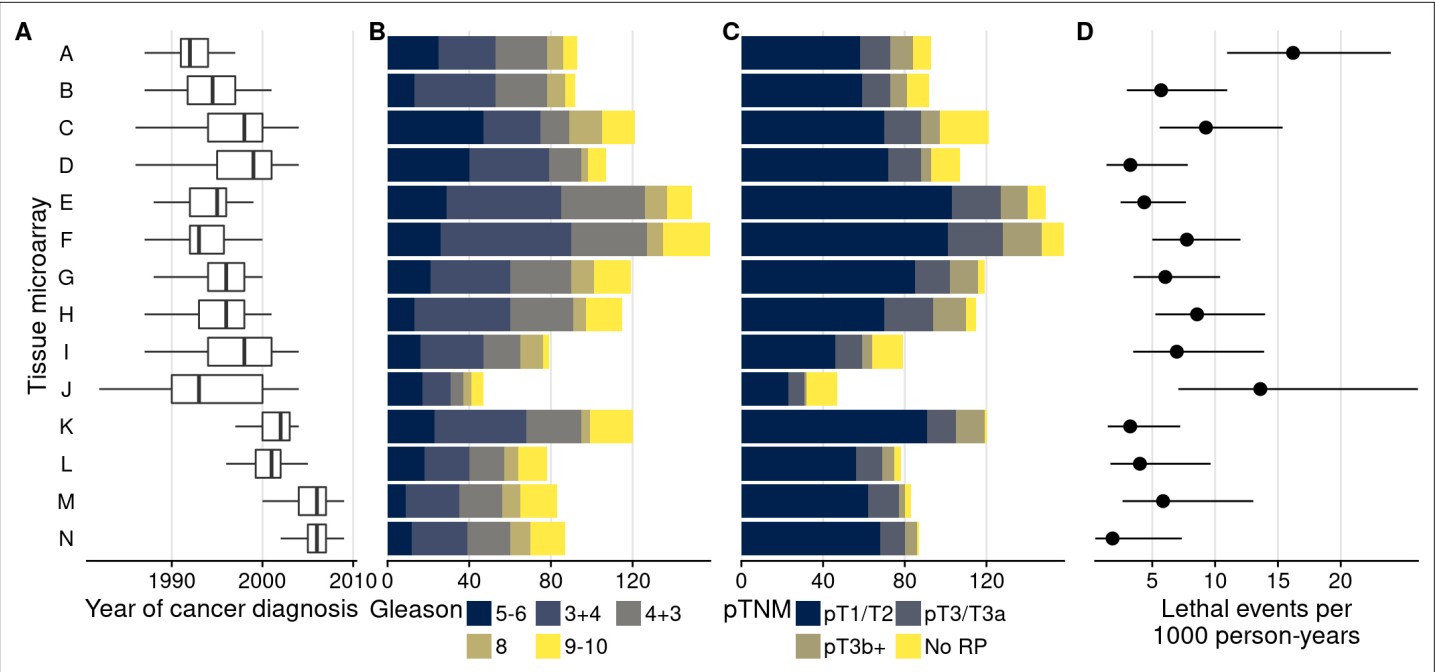

**Figure 1.** Characteristics of men with prostate cancer with tissue included on the 14 tumor tissue microarrays. (**A**) Calendar years of cancer diagnosis, with thick lines indicating median, boxes interquartile ranges, and whiskers 1.5 times the interquartile range. (**B**) Counts of tumors by Gleason score. (**C**) Counts of tumors by pathological TNM stage (RP: radical prostatectomy). (**D**) Rates of lethal disease (metastases or prostate cancer-specific death over long-term follow-up), with bars indicating 95% confidence intervals. As throughout, multiple cores are summarized per tumor.

be expressed in more aggressive tumors with activation of the PI3K signaling pathway while PTEN expression would be expected to be low in the same tumors (*Stopsack et al., 2020*).

Further supporting the second hypothesis, we did not observe any meaningful reduction in ICCs when we considered tumors that had the same clinical features in terms of Gleason score and stage (*Figure 4—figure supplement 2*). Moreover, the association between Gleason score and biomarker levels (*Figure 2D*) was considerably lower than between TMAs and biomarker levels, as underscored by less pronounced visual separation of principal components by Gleason score (*Figure 4C*) than by TMA (*Figure 4B*). Gleason score differences explained no more than 13% of the variance in biomarker levels (for prostate-specific membrane antigen, PSMA; 95% CI for ICC, 0.02–0.29), and 13 of the 20 biomarkers had ICCs by Gleason score of 1% or less (*Figure 4—figure supplement 3*).

To directly disentangle both hypotheses, we further examined data on 10 tumors with a total of 53 tumor cores for which some cores were included on different TMAs (*Figure 4D*). These were not included in the previous analyses and had estrogen receptor scoring data. This design allowed us to estimate biomarker differences directly attributable to between-TMA variability within the same tumors while controlling for the between-core variability expected due to intratumoral heterogeneity. Of the total variance in estrogen receptor alpha levels, 30% (95% CI, 0–67) was explained by between-TMA variation; for estrogen receptor beta, 24% (95% CI, 0–60) was explained by between-TMA variation. For comparison, between-tumor variation explained 37% (95% CI, 4–68) of the variance of estrogen receptor alpha levels and 26% (95% CI, 0–57) of the variance of estrogen receptor beta levels.

Collectively, while these observations highlighted moderate differences in clinical and pathological characteristics between TMAs, they suggested that between-TMA differences were largely due to batch effects.

## Mitigation of batch effects

We implemented six different approaches for batch effects mitigation and compared these to the uncorrected biomarker levels (*Figure 3*, *Figure 3—figure supplement 1*). Two mitigation approaches, batch means (approach 2) and quantile normalization (approach 6), assumed no true difference between TMAs is arising from patient and tumor characteristics, while all other approaches attempted

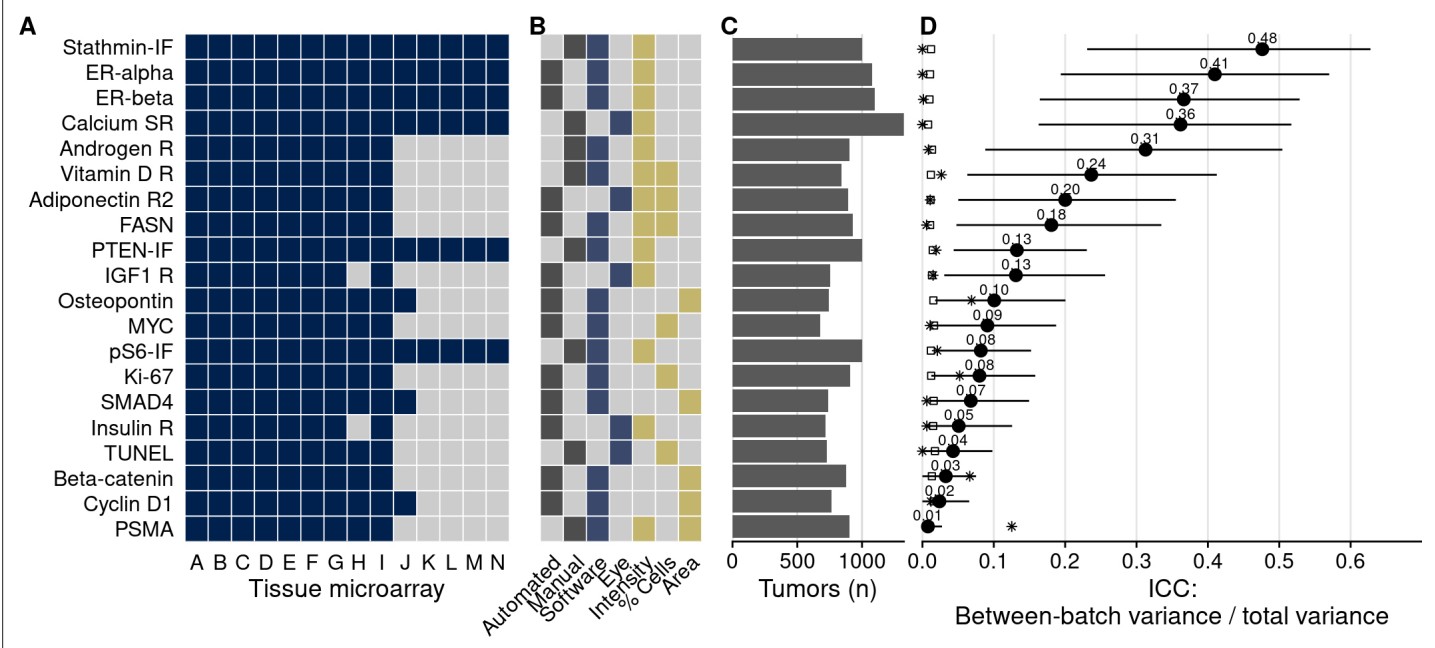

**Figure 2.** Biomarkers stained, staining and scoring methods, and intraclass correlation coefficients (ICCs). (**A**) Tissue microarrays (TMAs) assessed for each marker (dark blue, yes). (**B**) Approach to staining biomarkers: automated staining system versus manual staining (gray, yes); quantification of biomarkers: software-based scoring versus eye scoring (blue, yes); biomarker quality assessed: staining intensity, proportion of cells positive for the biomarker, area of tissue positive for the biomarker (yellow, yes). (**C**) Counts of tumors assessed for each biomarker. (**D**) Between-TMA ICCs (i.e., proportion of variance explained by between-TMA differences) for each biomarker, with 95% confidence intervals. Empty symbols indicate the 97.5th percentile of the null distribution of the ICC obtained by permuting tumor assignments to TMAs; asterisks indicate between-Gleason grade group ICCs. Biomarkers are arranged by descending between-TMA ICC.

The online version of this article includes the following figure supplement(s) for figure 2:

**Figure supplement 1.** Tissue microarrays (TMAs) and differences in % positivity, at the example of estrogen receptor alpha and beta, and variance in biomarker levels explained by between-TMA differences (ICC).

to retain such differences between TMAs. It is possible that the choice of mitigation approaches may be optimized using knowledge of the source of the batch effect. This would be the case if each method "specialized" in mitigating effect from specific sources. We have not investigated this possibility here. Overall, correlations between values adjusted by different approaches were higher (mean Pearson r, 0.97–1.00) than between uncorrected values and corrected values (r, 0.90–0.95), regardless of mitigation approach (*Figure 4E*).

Approaches 2–7 reduced visible separation by batch on plots of the first two principal components (*Figure 4—figure supplement 4*). Variance attributable to between-TMA differences decreased to ICCs of <1% for all markers (*Supplementary file 1b*). An exception was the quantile regression-based approach 5; the ICCs after this approach remained up to 10%. This method does not explicitly address differences in means between batches but allows associations between clinical and pathological factors and biomarker levels to differ at high and low quantiles (*Figure 4—figure supplement 5*).

The differences between uncorrected values and batch effect-corrected values were remarkably similar between the mean-based approaches using approaches 2 (simple means), 3 (standardized batch means), and 4 (inverse probability-weighted batch means; *Figure 4—figure supplement 6*). Consequently, batch effect-corrected values by approaches 2–4 were highly correlated (*Figure 4E*). All mean-only batch effect mitigations also gave the same results when fitting outcome models stratified by batch (*Figure 4—figure supplement 7*). However, batch-specific results differed for approaches that targeted between-batch differences in the variance of biomarkers.

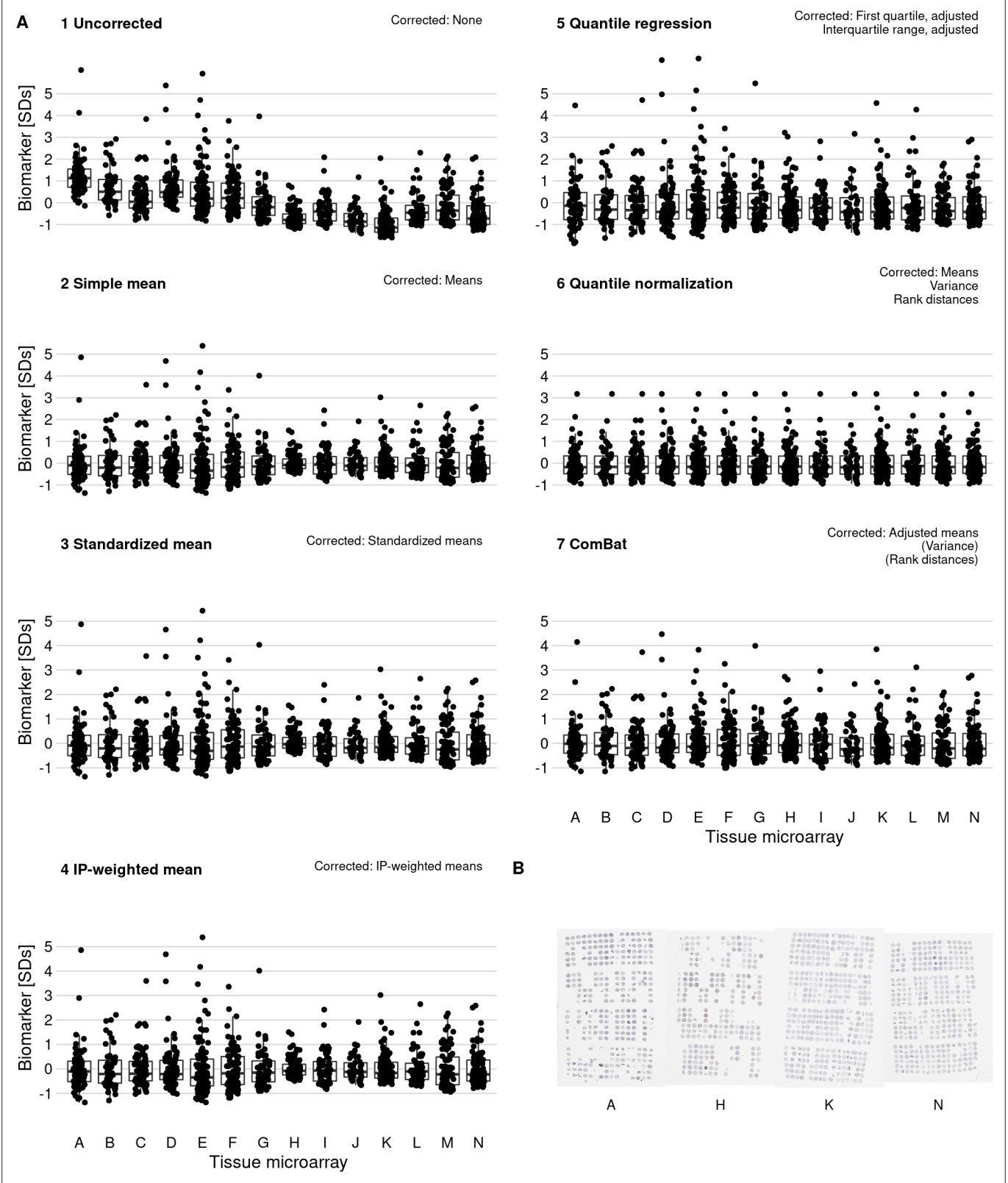

**Figure 3.** Effect of batch effect mitigation on a biomarker with strong between-tissue microarray (TMA) variation. (**A**) The protein biomarker estrogen receptor-alpha was quantified as staining intensity in nuclei of epithelial cells, averaged over all cores of each tumor. Each panel shows processed data for a specific approach to correcting batch effects. Notes in the upper right corner indicate which properties of batch effects were potentially addressed. Each data point represents one tumor. y-axes are standard deviations of the combined data for the specific method. Thick lines indicate medians, boxes

*Figure 3 continued on next page*

*Figure 3 continued*

interquartile ranges, and whisker length is 1.5 times the interquartile range. (**B**) Example photographs of TMAs; brown color indicates positive staining.

The online version of this article includes the following figure supplement(s) for figure 3:

**Figure supplement 1.** Uncorrected compared with batch effect-corrected biomarker levels for estrogen receptor alpha.

## Validating batch effect mitigation in plasmode simulation

To compare the performance of the different batch mitigation approaches in a time-to-event analysis, we applied plasmode simulation (*Franklin et al., 2014*) to fix the expected strength of the biomarker exposure–outcome relationship a priori before artificially introducing batch effects. The correlation structure between biomarker and confounders and between confounders and batches from the actual data (*Figure 5—figure supplement 1A, C*) was preserved in the plasmode-simulated data. Likewise, across a range of hazard ratios for the biomarker–outcome association, confounder–outcome associations remained unchanged (*Figure 5—figure supplement 1B, D*).

We first evaluated a setting in which we did not introduce batch effects (*Figure 5A*). Here, the observed hazard ratios without batch effect mitigation equaled the expected. When performing (unnecessary) batch effect mitigation, observed hazard ratios were still comparable with the expected hazard ratios (*Figure 5D*; see *Supplementary file 1c* for CIs).

We then introduced batch effects by adding batch-specific mean differences to the observed biomarker levels, yet without introducing differences in variance by batch (*Figure 5B*). Without batch effect mitigation, for a true hazard ratio of 3.0, the observed hazard ratio, averaged over simulations, was 2.17 (95% CI, 1.86–2.53), an underestimate by 28% (*Figure 5E*; *Supplementary file 1c*). In contrast, all mitigation approaches produced CIs that covered the expected hazard ratio (e.g., approach 6 quantile normalization: hazard ratio, 3.03; 95% CI, 2.48–3.69).

When we introduced batch-specific differences in both means and in variances (*Figure 5C*), the observed hazard ratio without batch effect mitigation decreased to 1.90 (95% CI, 1.66–2.16) compared to the expected hazard ratio of 3.0 (*Figure 5F*; *Supplementary file 1c*). Batch effect mitigation methods that only focus on means (approaches 2–4) reduced but did not fully eliminate bias, with hazard ratios ranging between 2.67 and 2.70. Methods that address differences in both mean and variance resulted in less bias, with an observed hazard ratio of 3.11 (95% CI, 2.54–3.81) for approach 6 (quantile normalization).

We also included two stratification-based approaches. Fitting survival models separately by batch, followed by inverse-variance pooling (approach 8), resulted in approximately unbiased estimates but was less efficient than other approaches, comes with a risk of sparse-data bias, and resulted in considerably wider CIs in our simulation. Including batch as a stratification variable in a single Cox model (approach 9) was unbiased and efficient. A drawback of both stratification-based approaches is that they do not explicitly estimate batch effect-adjusted biomarker values that could be visualized directly.

Scenarios evaluated thus far were based on the actual, modest imbalance of confounders between batches and at most weak associations between the biomarker and confounders, resulting in weak confounding overall. We additionally introduced both modest and strong associations between biomarker and confounders and created more severe imbalance between batches (*Figure 5—figure supplement 2*). In all scenarios, the ranking of mitigation methods was preserved (*Figure 5—figure supplement 3*, *Supplementary file 1c–d*), with the least bias obtained through quantile normalization (approach 6). Bias occurred when using uncorrected biomarker levels in the presence of any batch effects, except if there was no association between biomarker and outcome (i.e., a hazard ratio of 1), and with mean-only approaches 2–4 if variance was also affected by batch effects. In no situation, except possibly with the quantile regression-based approach 5, were estimates after batch effect mitigation farther from the expected values than results based on uncorrected biomarker levels.

## Impact of batch effects

To illustrate how batch effect mitigations alter the results of commonly conducted tumor biomarker analyses, we estimated how uncorrected and corrected biomarker levels were associated with Gleason score and with rates of lethal disease. For markers with little between-TMA variability (low ICCs) such as beta-catenin, there were no noticeable differences in associations between using unadjusted and adjusted biomarker levels irrespective of adjustment model, as expected from plasmode simulation.

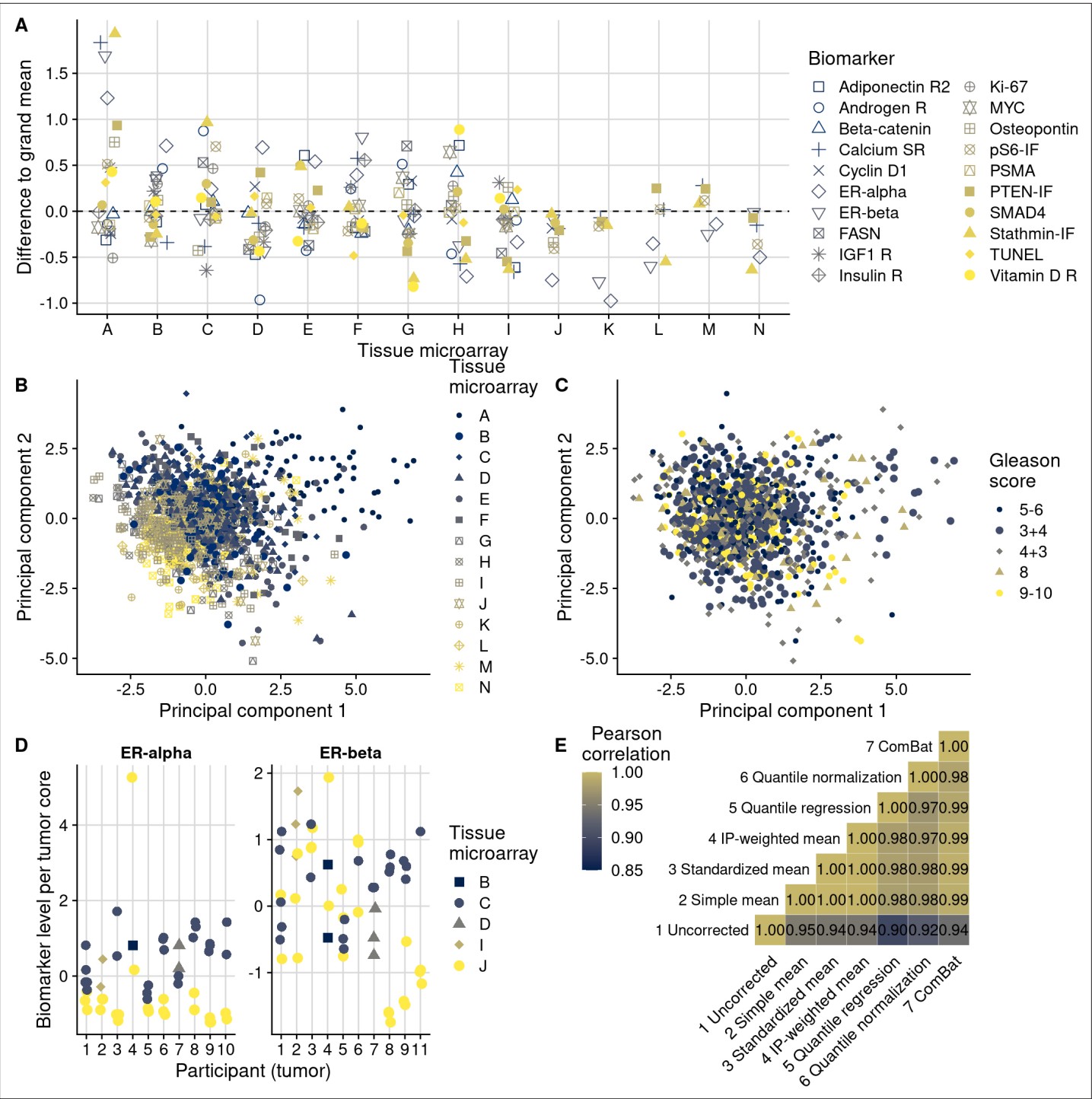

**Figure 4.** Patterns, source, and remediation of batch effects. (**A**), Biomarker mean levels by tissue microarray (TMA), in biomarker-specific standard deviations (y-axis). Each point is one TMA. (**B**) First two principal components of biomarkers levels on all 14 TMAs, with color/shape denoting TMA. Each point is one tumor. (**C**) The same first two principal components, with color/shape denoting Gleason score. (**D**) Per-core biomarker levels for tumors with multiple cores included on two separate TMAs, for estrogen receptor (ER) alpha and beta, both in standard deviations. Each point is one tumor core. (**E**) Pearson correlation coefficients r between uncorrected and corrected biomarker levels. Entries are averages across all markers.

The online version of this article includes the following figure supplement(s) for figure 4:

**Figure supplement 1.** Ratios of variance per tissue microarray to the mean variance for each marker.

**Figure supplement 2.** Intraclass correlation coefficients (ICCs), quantifying the proportion of variance in biomarker levels attributable to between-tissue microarray (TMA) differences, across all tumors and after restriction to those 378 tumors across TMAs that have the same clinical/pathological

*Figure 4 continued on next page*

*Figure 4 continued*
characteristics in terms of Gleason score 3+4 and prostatectomy stage pT1/T2.

**Figure supplement 3.** Intraclass correlation coefficients (ICCs), quantifying the proportion of variance in biomarker levels attributable to between-Gleason grade differences, sorted by increasing ICC.

**Figure supplement 4.** Principal components 1 and 2 after batch effect correction using quantile normalization (approach 6) for biomarkers available on all tissue microarrays (TMAs).

**Figure supplement 5.** Quantile-specific associations of confounders (clinical/pathological differences) with (uncorrected) biomarker levels of estrogen receptor alpha.

**Figure supplement 6.** Batch corrections per tissue microarray and method.

**Figure supplement 7.** Biomarker differences in ER-alpha intensity, after batch effect correction methods, for a one-unit increment in Gleason score, stratified by tissue microarray (TMA).

However, for markers with higher between-TMA variability (higher ICC) and stronger associations with the outcome, adjustment approaches led to noticeable differences (*Figure 6*). For example, uncorrected stathmin expression levels were not associated Gleason score (difference, 0.00 standard deviations per one grade-group increase; 95% CI, −0.05 to 0.05), while the difference in levels corrected according to approach 6 (quantile normalization) was 0.04 (95% CI, 0.00 to 0.07), suggesting a potentially qualitatively different interpretation (*Figure 6A*; *Supplementary file 1e*). In models for lethal disease (*Figure 6B*), the otherwise unadjusted hazard ratio for the highest quartile of the vitamin D receptor, compared to the lowest quartile, was 0.44 (95% CI, 0.23–0.86); after mitigation using approach 6, the hazard ratio was 0.19 (95% CI, 0.09–0.40), suggesting that unadjusted biomarker levels could underestimate the prognostic association by 2.3-fold (*Supplementary file 1f–g*).

## Discussion

The key strength of using TMAs is their utility in parallelizing the assessment of biomarkers on a large number of tissue specimens (*Kononen et al., 1998*). Similar to other high-throughput platforms, batch effects have to be considered in every TMA biomarker study. As we demonstrated, for some of the biomarkers, batch effects can be of substantial magnitude. We show that batch effect mitigation is possible and can enhance study findings.

In our study of prostate tumor specimens, between-TMA differences explained 10% or more of the variance in biomarker levels for half of the included biomarkers, considerably more than one of the strongest pathological features in prostate cancer, Gleason grade. All analytical mitigation approaches to reduce batch effects, whether they attempted to retain real differences between tumors from different TMAs or not, led to corrected biomarker levels that were more similar to each other than they were, in general, to the uncorrected biomarker levels. In drawing from a large set of protein tumors biomarkers in prostate cancer, we show how appropriately mitigating batch effects strengthens results and their validity for biomarkers affected by batch effects.

Ideally, batch effects between TMAs are minimized when designing a study. Standardizing how tumor samples are obtained, stored, processed, and assayed is critical, as are stratified or random allocation of tumor samples to different TMAs (*Tworoger and Hankinson, 2006*) when possible. However, the batch effects that we observed occurred despite all feasible standardization efforts. Moreover, samples will be collected sequentially, and TMAs may be constructed sequentially in large-scale prospective studies over time. There were modest differences in the clinical and pathological characteristics between our TMAs, an issue that may be inevitable in larger-scale biobank studies. Allocation schemes of tumors to TMAs that appear ideal retrospectively, for example by matching "cases" of lethal tumors with "controls" of non-lethal tumors, may not be feasible prospectively. Likewise, in few of the thousands of studies using TMAs will it be possible to reallocate tumors to different TMAs and repeat all pathology work merely to reduce the implications of batch effects.

An additional challenge in the design phase is that tissue samples are inherently heterogeneous and cannot simply be diluted, like blood samples. "Quality control" tumor samples that could serve as a quantitative calibration series suitable for all future biomarkers do not exist. One potential strategy is to include cell lines that have been formalin-fixed and paraffin-embedded on each TMA. While cell lines address issues of heterogeneity, the cell lines are often genomically unique and as such may not

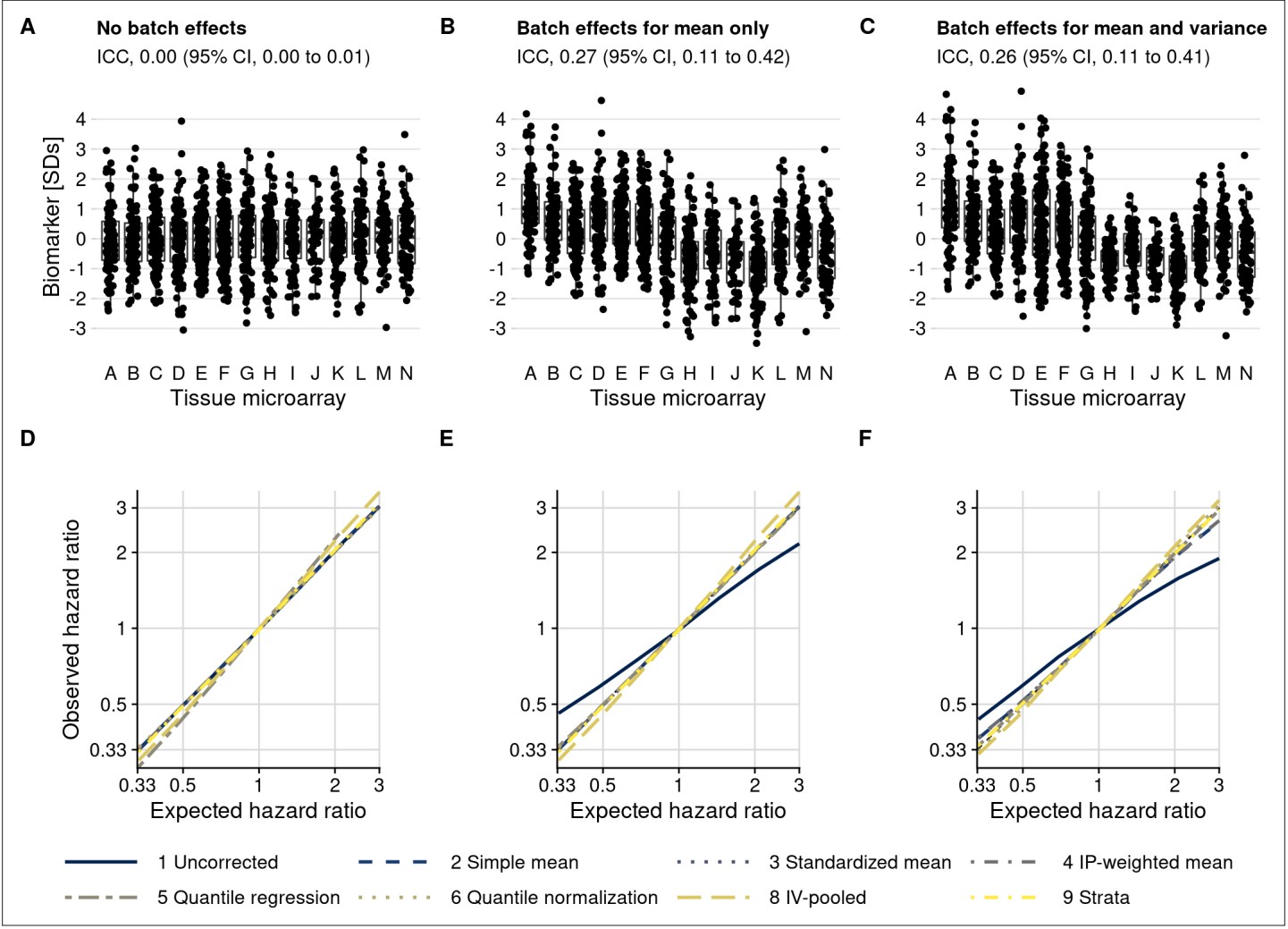

**Figure 5.** Plasmode simulation results. (**A–C**) Biomarker levels by tissue microarray in three simulation scenarios. (**D–F**) True versus observed hazard ratios for the biomarker–outcome association after alternative approaches to batch effect correction, with correction methods being numbered as in the Materials and methods section. The solid blue line indicates no correction for batch effects. (**A**, **D**) no batch effects; (**B**, **E**) means-only batch effects; (**C**, **F**) means and variance batch effects.

The online version of this article includes the following figure supplement(s) for figure 5:

**Figure supplement 1.** Data structures in the actual data and in 200 plasmode simulation data sets.

**Figure supplement 2.** The data correlation structure "confounding and imbalance."

**Figure supplement 3.** Plasmode simulation results for all scenarios.

be relevant for all biomarkers. Another potential approach is to include samples from the same tumor case across TMAs, which would allow for direct estimation of batch effects. For these reasons, a principled approach that anticipates batch effects and addresses them analytically is critical.

Beyond efforts to prevent batch effects during the study design phase, we suggest the following best practices when undertaking TMA-based tissue biomarker studies (*Figure 7*). First, the extent of potential batch effects should be explored and reported in any study of cancer tissue using TMAs. Inspecting TMA slides and plots (*Figure 3*; *Manimaran et al., 2016*) is important. Between-TMA variation should be quantified, for example by calculating ICCs, that is, to contrast variation of biomarker levels between TMAs compared to that between or within tumors (*Nakagawa and Schielzeth, 2010*). In our study, for half of the biomarkers, ICCs for between-TMA variation were low, at less than 10%, although the proportion of tolerable batch variation should be chosen based on the context. Whether TMAs differ in terms of average biomarker levels, low levels (possibly reflective of background), or variability between tumors will also inform what impact of between-TMA differences to expect.

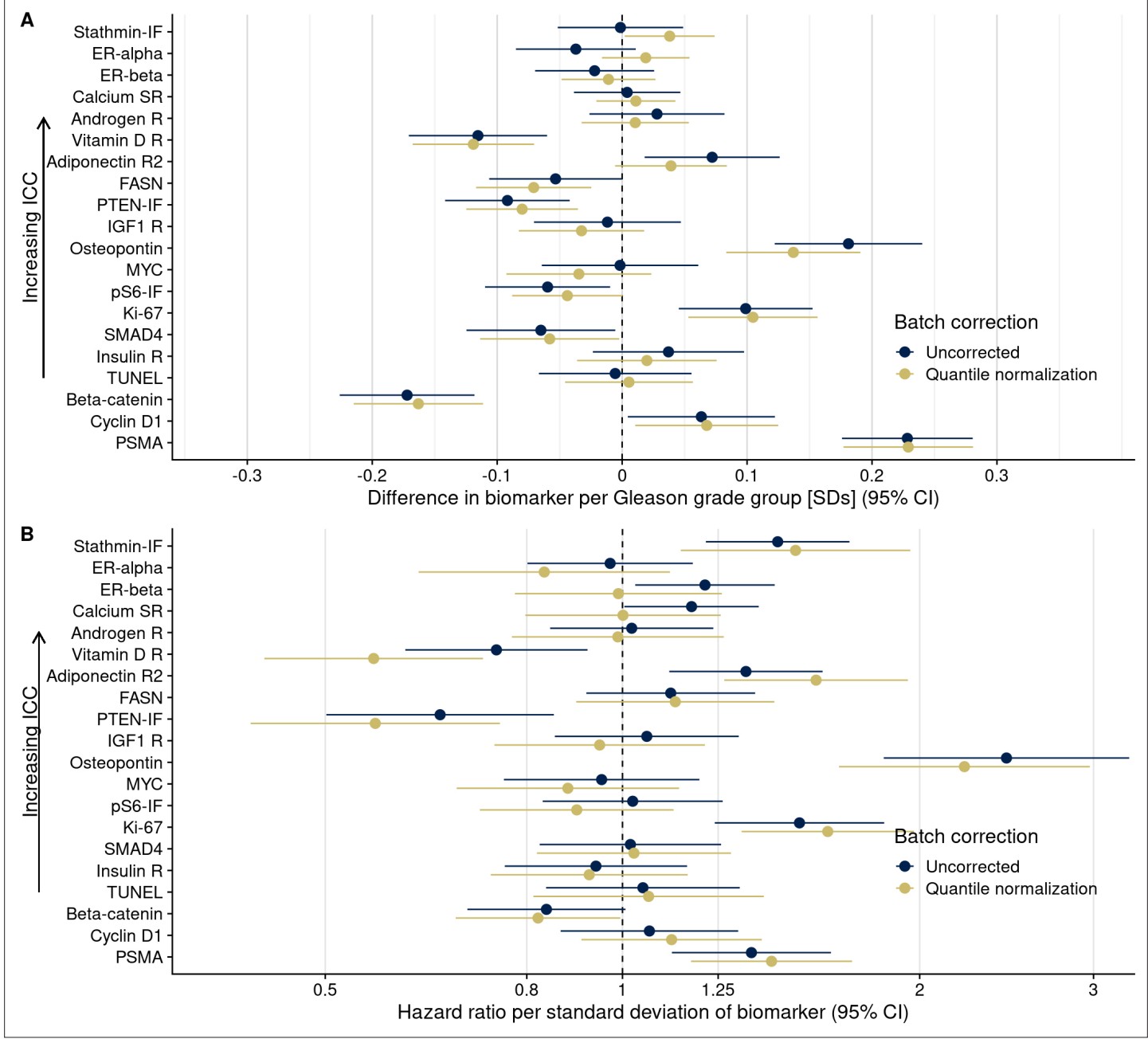

**Figure 6.** Consequences of batch effect mitigation on scientific inference. (**A**) Gleason score and biomarker levels (in standard deviations, per Gleason grade group). (**B**) Biomarker levels and progression to lethal disease, with hazard ratios per one standard deviation increase in biomarker levels from univariable Cox regression models. In both panels, blue dots indicate estimates using uncorrected biomarker levels, yellow dots indicate batch effect-corrected levels, applying quantile normalization (approach 6). Lines are 95% confidence intervals. Biomarkers are ordered by decreasing between-tissue microarray intraclass correlation coefficient (ICC).

Second, the source of between-TMA differences should be elucidated. Ideally, including multiple cores from the same tumors in more than one TMA will help estimating, again using ICCs, how biomarker levels vary between TMAs, between tumors, and within tumors. Alternatively, ICCs between TMAs can be estimated by restricting to or adjusting for tumor features associated with differences in the biomarker, if known. In our study, both approaches indicated that the largest share of between-TMA differences was likely due to batch effects rather than due to true differences between tumors on different TMAs. However, one should not simply assume this to be the case in other settings, and also explore between-tumor differences as one source of between-TMA differences.

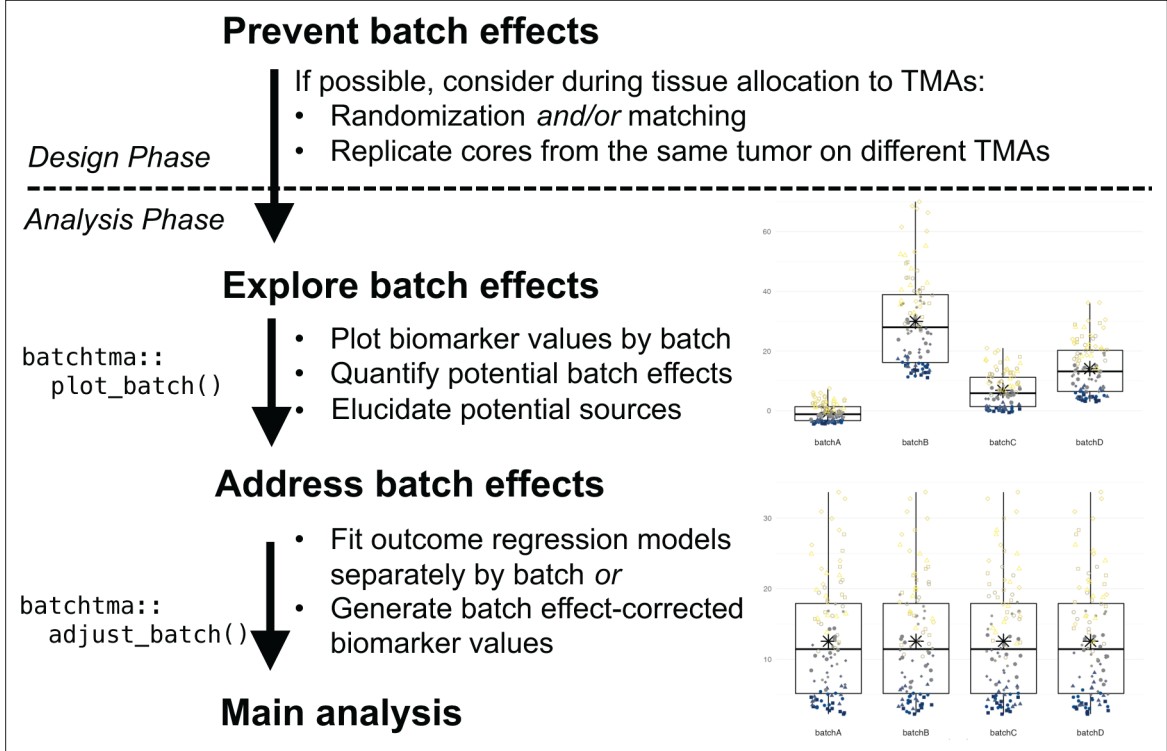

**Figure 7.** Recommended workflow for anticipating and handling batch effects between tissue microarrays (TMAs). Following prevention approaches at the design phase of a project, all TMA-based studies should explore the potential for batch effects once a biomarker has been measured. Addressing batch effects should only be skipped there is no indication for their presence. Batch effect-corrected biomarker levels can easily be generated by the *batchtma* R package.

In multidisciplinary team discussions (***Marrone et al., 2019***), it may be possible to directly pinpoint the source of batch effects and eliminate its cause. All study steps, including the pre-analytic, analytic, and post-analytic phases, should be considered. If sources of batch effects can be identified, it is preferable that they be addressed directly during the pre-analytical or analytical phase, rather than applying the post-analytical methods that we have described here and that may not adequately incorporate knowledge on the source of batch effects. For example, if immunohistochemical staining was performed separately for each TMA, then immunohistochemistry and quantification should be repeated using new sections from all TMAs at once. Imaging of pathology slides can also be a source of batch effects (***Kothari et al., 2014***), as could be image analysis. In other cases, particularly if such obvious reasons for batch effects were avoided through standardized processing, as in our examples, it may remain elusive whether batch effects were induced through subtle differences in how tumors were cored and embedded during TMA construction, how long they had been stored, how they were sectioned, how well the staining process was standardized, or how successfully background signal was eliminated during software-based quantification. Yet even biomarkers scored by manual quantification were not free from batch effects.

Third, if a biomarker is affected by batch effects and no "physical" remediation is possible, then post-analytical approaches should be used to reduce bias in results (***Tworoger and Hankinson, 2006***; ***Leek et al., 2010***). We demonstrate that in all plausible or exaggerated real-world scenarios, estimates after applying batch effect mitigations were consistently closer to the true underlying values than they were without. If batches do not only differ in terms of mean values, but also in terms of their variances, then methods that focus solely on means may be insufficient. A simple quantile-normalization-based approach was successful in reducing bias in real-world scenarios and could be preferred for its simplicity. It is important to note that any method tested in this study is preferable over not addressing batch effects, and thus the choice between methods should be secondary to the choice to address batch effects altogether. Only results for biomarkers that are affected by batch effects and that are associated with the outcome of interest will show large changes in estimates, as

the vitamin D receptor in our example. In contrast, for the majority of our example biomarkers, results did not change appreciably because batch effects were low, associations with the outcome were close to null, or both (*Figure 6*).

We recommend that researchers openly address batch effects in their TMA-based studies: they are not an error of an individual study, but an inherent feature of TMA-based studies. Batch effects have long been recognized in studies of the transcriptome using microarrays and next-generation sequencing, where batch effect mitigations are a component of standard workflows (*Leek et al., 2010*; *Leek et al., 2012*). Our data strongly suggest that protein biomarker studies using multiple TMAs are at risk of batch effects just like any other biomarker study. The extent of batch effects is difficult to predict, and empirical evaluation is necessary each time. Future studies should quantify between-TMA differences and, if they deem batch effect mitigations to be unnecessary, provide evidence for absence of batch effects, rather than merely assuming their absence. The methods that we provide facilitate the appropriate migration of batch effects between TMAs and help strengthen scientific inference. It may be prudent to extend this approach to in-situ tissue biomarkers other than proteins, such as RNA in-situ hybridization, even if our study only demonstrated batch effects for proteins. Having mitigated batch effects will allow researchers to focus on increasing study validity by addressing other sources of measurement error (*van Smeden et al., 2020*), selection bias (e.g., from tumor biospecimen availability) (*Liu et al., 2018*), and confounding.

## Materials and methods
### TMAs and biomarkers
Tumor tissue in this study was from men who were diagnosed with primary prostate cancer during prospective follow-up of two nationwide cohort studies. The Health Professionals Follow-up Study is an ongoing cohort study that enrolled 51,529 male health professionals across the United States in 1986. The Physicians' Health Study 1 and 2 were randomized-controlled trials of aspirin and dietary supplements, starting in 1982 with 22,071 male physicians. Participants were diagnosed with and treated for prostate cancer at local health care providers across the United States. The study team collected formalin-fixed paraffin-embedded tissue specimens from radical prostatectomy and transurethral resection of the prostate (TURP), and study genitourinary pathologists performed central re-review, including standardized Gleason grading of full hematoxylin–eosin-stained slides from the tumor blocks (*Stark et al., 2009*). Written informed consent was obtained from all participants, and the study protocol was approved by the institutional review boards of the Brigham and Women's Hospital and Harvard T.H. Chan School of Public Health (IRB19-1430), and those of participating registries as required.

TMAs were constructed using 0.6 mm tissue cores of the primary nodule or the nodule with the highest Gleason score (*Pettersson et al., 2012*), including three or more cores of tumor tissue per participant (tumor). For a subset of tumors, additional cores of tumor-adjacent, histologically normal-appearing prostate tissue were included. TMAs were constructed at the same laboratory across a 10-year period, as tissue from cohort participants became available, without matching on patient or tumor characteristics and without randomization. Cores from the same participant were generally included on the same TMA, with exceptions noted below, and summarized as the mean. We include information from 14 prostate tumor TMAs.

Immunostaining was generally performed separately for individual biomarkers yet always for all TMAs at the same time. Detailed immunohistochemistry staining and quantification procedures for each marker have been published (*Rider et al., 2015*; *Flavin et al., 2014*; *Fiorentino et al., 2008*; *Ahearn et al., 2016*; *Ding et al., 2011*; *Nguyen et al., 2010*; *Pettersson et al., 2018*; *Kasperzyk et al., 2013*; *Dhillon et al., 2009*; *Hendrickson et al., 2011*; *Zu et al., 2013*; *Stopsack et al., 2020*) or are in preparation for estrogen receptor alpha (antibody SP1; Thermo Fisher Scientific, Waltham, MA) and an antibody (PPG5/10; Bio-Rad Laboratories, Hercules, CA) widely used to measure estrogen receptor beta. If batch effect mitigation approaches had been applied in the original studies, the uncorrected levels were retrieved. Right-skewed biomarker scores (Ki-67, pS6, TUNEL) were $\log_e$ transformed. All biomarkers were scaled to mean 0 and standard deviation 1 solely to facilitate comparisons of batch effects across markers; batch effect mitigation does not necessitate scaling and preserves absolute biomarker values.

## Extent and type of batch effects

To visualize the extent of biomarker variation between TMAs, we plotted uncorrected biomarker values by tumor, biomarker, and TMA. We summarized biomarker variation using the first two principal components (*Lê et al., 2008*). We calculated between-TMA mean differences and ratios of variances versus the first TMA. We tested if tumors with different clinical/pathological characteristics had higher biomarker levels in TMAs with higher means (i.e., multiplicative effect modification). For each biomarker and each clinical/pathological feature (ordinal Gleason score, ordinal stage, or calendar year of diagnosis), let $Z_{ij}$ be the within-TMA z-score (mean 0, standard deviation 1) for tumor $i$ from TMA $j$; $A_i$, the clinical/pathological feature of tumor $i$; $B_j$, the TMA-specific biomarker mean, $r_j$, the TMA-specific random effect, and $e_{ij}$, residual error. In the regression model $Z_{ij} = \beta_0 + \beta_1 A_i + \beta_2 B_j + \beta_3 A_i B_j + r_j + e_{ij}$, we evaluated the $\beta_3$ term to assess for multiplicative effect measure modification.

We calculated the proportion of variation in biomarker levels attributable to TMA using intra-class correlations (ICCs, also "repeatability"; *Nakagawa and Schielzeth, 2010*) based on one-way random effects linear mixed models with an independent variance–covariance structure (*Nakagawa and Schielzeth, 2010*; *Hankinson et al., 1995*) for $Y_{ij}$, the biomarker level per tumor $i$ and TMA $j$; where $\beta_0$ is the biomarker mean; $r_j$, the random effect for TMA $j$; and $e_{ij}$, the residual error: $Y_{ij} = \beta_0 + r_j + e_{ij}$. The ICC was defined as the proportion of between-TMA variance in the total variance: $ICC = \frac{var(r)}{var(r)+var(e)}$. 95% CIs for ICCs were obtained using parametric bootstrapping using 500 repeats (*Stoffel et al., 2017*).

## Source of batch effects

To directly distinguish between-TMA variation caused by batch effects from variation caused by differences in patient and tumor characteristics, we compared ICCs per biomarker overall to ICCs per biomarker when restricting analyses to a subset of tumors with the same clinical features. We also leveraged a small subset of tumors that had cores included on more than one TMA. Here, we used two-way random effects linear mixed models with independent variance–covariance structure (*Bates et al., 2015*) to separate between-TMA variation from between-core variation (i.e., intratumoral heterogeneity) and residual modeling error: $Y_{ijk} = \beta_0 + r_j + s_i + e_{ijk}$. Compared to the model described earlier, this model additionally includes tumor-specific random effects $s_i$, and thus $ICC = \frac{var(r)}{var(r)+var(s)+var(e)}$.

## Mitigation of batch effects

In addition to (1) using uncorrected values, we implemented eight different approaches to handle between-TMA batch effects:

(2) Simple means. This approach assumes that all TMAs, if not affected by batch effects, would have the same mean biomarker value. Differences in mean biomarker values per batch are corrected by estimating batch-specific mean effects (differences from the overall mean level) using a linear regression model with uncorrected biomarker values as the outcome and batch indicators as predictors. Corrected biomarker values are then obtained by subtracting batch-specific effects from the uncorrected biomarker values. Mean differences can either indicate the difference of each batch mean to the overall mean, as implemented here, or be defined by comparison to a reference batch.

(3) Standardized means. This approach estimates marginal means per batch using model-based standardization (in the epidemiologic sense). It assumes that batches with similar characteristics have the same means if not affected by batch effects. A linear regression model for a specific biomarker is fit, adjusting for tumor variables that differ in distribution between TMAs, similar to an approach described in the epidemiology literature by *Rosner et al., 2008*. Let $Y_{ij}$ indicate the biomarker value for tumor $i$ on TMA $j$; $B_j$, TMA $j$; $C_1$ to $C_m$, the m covariates to be retained; and $e_{ij}$, the residuals. Then $Y_{ij} = \beta_0 + \beta_j B_j + \gamma_1 C_1 + \ldots + \gamma_n C_n + e_{ij}$. Batch effect-corrected biomarker values can be obtained by subtracting batch-specific effects $\beta_j$ predicted from the model above from uncorrected biomarker values.

We included the following clinical and pathologic variables as plausible sources of real between-TMA differences that should be retained in this approach, as well approaches 4–7: calendar year of diagnosis (linear), Gleason score (categorical: 5–6; 3+4; 4+3; 8; 9–10), and pathologic tumor stage (categorical: pT1/T2, pT3/T3a, pT3b/T4/N1, missing/tissue from transurethral resection of the prostate).

(4) Inverse-probability weighted batch means. Like the preceding approach, this approach assumes that batches with similar characteristics have the same means if not affected by batch effects. While the preceding approach assumes a constant association between covariates and biomarker levels across batches, this approach allows for associations to differ between batches. We first used inverse probability weighting for marginal standardization of the distribution of clinical and pathological features per batch to the distribution in the entire study population. Stabilized weights (*Cole and Hernán, 2008*), truncated at the 2.5th and 97.5th percentile, were obtained through multinomial regression models, modeling the probability of assignment to a specific batch based on same clinical and pathological variables as in approach 3. In the weighted pseudopopulation, we then used a marginal linear model to estimate batch-specific mean differences, which were further used as in approaches 2 and 3.

(5) Quantile regression. This approach assumes that batches with similar characteristics have the same values for a selected set of batch-specific quantiles, in this application the upper and lower quartile. The lower quartile may be particularly affected by background noise, while the upper quartile may more likely reflect differences in batches due to covariates. A corollary of separately modeling the two differently is that clinical and pathological variables are allowed to have different effects on these quartiles (*Bann et al., 2020*). These assumptions contrast with approaches 2–4 that focus on mean levels only. We used quantile regression with the Frisch-Newton approach (*Portnoy and Koenker, 1997*) separately for the first and third quartile of biomarker values with batch indicators to predict adjusted batch-specific quantile values with the same confounders as above. We then used the batch-specific 25th percentiles ($y^{\tau=0.25}$) as the offset and the interquartile range between the 25th and 75th percentiles ($y^{\tau=0.75}$) as the scaling factor when batch-correcting biomarker levels. Let $y_{ij}$ indicate the batch effect-corrected biomarker level for tumor i on TMA j; $y_{ij}$, the uncorrected biomarker level for tumor i on TMA j; $y_i^{\tau=x}$, xth quantile of y for batch j (predicted value for $y_j$ from unadjusted quantile regression); $\hat{y}_j^{\tau=x,*}$ is $\hat{y}_j^{\tau=x}$ with adjustment for confounders (predicted value for $y_j$ from adjusted quantile regression); and $y^{\tau=0.75}$, the xth quantile of y overall. Then the corrected biomarker level is

$$y_{ij}^* = \frac{(y_{ij}-\hat{y}_j^{\tau=0.25})(\bar{y}^{\tau=0.75}-\bar{y}^{\tau=0.25})}{(\hat{y}_j^{\tau=0.75,*}-\hat{y}_j^{\tau=0.25,*})} + \bar{y}^{\tau=0.25} - \hat{y}_j^{\tau=0.25,*} + \hat{y}_j^{\tau=0.25}$$

(6) Quantile normalization. This approach assumes that samples on all batches, if not affected by batch effects, would not only have the same mean and variance but also the same distribution of individual biomarker values. Uncorrected biomarker values are ranked within each batch and then ranks are replaced by the mean of values with the same rank across batches. We implemented quantile normalization using *limma* (*Ritchie et al., 2015*; *Bolstad et al., 2003*).

A conceptually related approach, for example, employed in molecular epidemiology (*Tworoger and Hankinson, 2006*; *Marrone et al., 2019*), would be to use within-batch ranks as the batch-corrected biomarker, often grouped into data-driven categories such as batch-specific quartiles. We did not further consider these derivatives because they do not retain absolute biomarker levels and can distort rank distances.

(7) ComBat. For comparison, we additionally included the ComBat algorithm, which like approach 4 attempts to retain differences in batch means due to covariate differences; it is frequently applied together with approach 6. ComBat and its derivatives (*Leek et al., 2012*; *Johnson et al., 2007*; *Zhang et al., 2018*) were initially designed for microarray studies of gene expression, which include considerably more than one biomarker per sample. This property would typically limit their use for a protein biomarker quantified on a TMA unless a large number of biomarkers is available, as in our study. Mitigation depends on values of other biomarkers on the same batches. Even if multiple protein biomarkers were available, the non-randomly selected set of concomitantly available biomarkers may influence how batch effects are corrected. ComBat scales means and (optionally) variances while (optionally) retaining adjustment variables. ComBat is implemented using an empirical Bayes approach to achieve more favorable properties for small batches. The underlying model is similar to the regression above and has been emulated by a two-way analysis of variance (*Nygaard et al., 2016*). In using ComBat, we scaled both means and variances, adjusting for the same clinical and pathological variables as before. Because ComBat cannot handle biomarkers if they are missing on entire batches, we ran ComBat separately for groups of biomarkers measured on 8, 9, 10, or 14 TMAs.

(8) Stratification with inverse-variance pooling. An alternative approach to treating batch effects is to estimate outcome regression models separately by batch. This approach can be applied for a

variety of regression models but does not result in corrected values. We pooled estimates with inverse variance-weighting to obtain summary estimates.

(9) Stratification in Cox proportional hazards regression. In a special case of stratification for time-to-event outcomes, Cox proportional hazards models allow for nonparametric batch effect mitigation by including batch as a stratification factor in the model specification. Comparisons are performed within batches. Unlike approach 8, only batch-specific baseline hazard functions but no batch-specific effects are estimated.

For approaches 1–7, we calculated Pearson correlation coefficients between uncorrected and corrected biomarker levels. Additionally, we repeated ICC and principal components analyses with corrected levels, and we estimated associations between Gleason score and biomarker levels after batch effect mitigation, stratifying by batch using approach 8.

Approaches 2–6, which result in batch effect-adjusted biomarker levels, are implemented in the R package *batchtma*, available at https://stopsack.github.io/batchtma and the Comprehensive R Archive Network (CRAN).

## Plasmode simulation

We evaluated the impact of batch effect mitigation approaches on known, investigator-determined biomarker–outcome associations using plasmode simulation, an approach used, for example, for evaluating confounding control for binary exposures in pharmacoepidemiology (*Franklin et al., 2014*). We used observed data from all tumors included on the 14 TMAs to determine covariates (Gleason grade, pathological stage) and outcome (lethal disease), preserving the observed correlation structure (e.g., joint distribution of clinical characteristics across TMAs). The only simulated elements were the biomarker levels and the strengths of biomarker–outcome associations (hazard ratios ranging from 1/3 to 3) that we fixed by simulating event times with flexible parametric survival models (*Crowther and Lambert, 2013*). Models used a baseline hazard function consisting of cubic splines with three knots (*Jackson, 2016*). Group differences were based on proportional hazards for the observed confounder–outcome coefficients in the real data and the fixed biomarker (exposure)–outcome hazard ratios.

First, we used plasmode simulation to generate the fixed associations of the biomarker levels with the outcome, which are unknowable outside simulation studies, generating 200 plasmode data sets for each association. Second, we introduced batch effects. Batch effects were either only for the mean or for both mean and variance, using the actual standardized between-TMA differences in means and variances for the estrogen receptor-alpha protein, a biomarker with high ICCs. We also added batch effects for mean and variance with effect modification, making mean and variance changes due to batch effects strongly correlated with Gleason scores. Third, we calculated batch effect-adjusted biomarker levels using approaches 2–6. Finally, we compared the expected hazard ratios for the biomarker–outcome association, fixed during simulations, with the estimated hazard ratios from Cox regression (with normality-based 95% CIs) before and after batch effect mitigation approaches 2–6 and using the two stratification-based approaches 8 and 9.

In sensitivity analyses, we simulated "moderate" associations between the biomarker and confounders (0.2 standard deviations difference in biomarker levels per Gleason grade group, 0.1 per stage category), "strong" associations (differences of 0.4 and 0.2 standard deviations, respectively; stronger than observed for any biomarker in our study), as well as "strong" associations and additional imbalance in Gleason grade and stage between TMAs (by excluding tumors with low grades from TMAs with higher-than-average means and excluding tumors with high stage from TMAs with low-than-average means), all before the four steps described above.

## Impact of batch effects

To quantify the impact of different approaches to batch-effect handling on scientific inference, we focused on two commonly employed types of analyses in biomarker research in prostate cancer: first, a cross-sectional analysis of Gleason score and biomarker levels, using linear regression models; second, a time-to-event analysis of biomarker levels and rates of lethal disease, using Cox proportional hazards regression. Gleason scores were modeled as ordinal variables and biomarkers as linear variables to obtain one single estimate per model. We also categorized biomarkers into four quartiles and compared hazard ratios for lethal disease of the extreme quartiles. Models were designed only for investigating issues of batch effects and not for subject matter inference on specific biomarkers.

## Data availability

The batchtma R package is available at https://stopsack.github.io/batchtma and the Comprehensive R Archive Network (CRAN). Code used to produce results this manuscript is at https://github.com/stopsack/batchtma_manuscript (copy archived at swh:1:rev:a588f10906f8685b055e5a6f0a487f5f850d13bc, *Stopsack, 2022*). Data are available for analysis on the Harvard FAS computing cluster through a project proposal for the Health Professionals Follow-up Study (https://sites.sph.harvard.edu/hpfs/for-collaborators).

## Acknowledgements

The authors thank the participants and staff of the HPFS and the PHS for their valuable contributions. In particular, the authors would like to recognize the contributions of Liza Gazeeva, Siobhan Saint-Surin, Robert Sheahan, Betsy Frost-Hawes, and Eleni Konstantis. The authors would like to thank the following state cancer registries for their help: AL, AZ, AR, CA, CO, CT, DE, FL, GA, ID, IL, IN, IA, KY, LA, ME, MD, MA, MI, NE, NH, NJ, NY, NC, ND, OH, OK, OR, PA, RI, SC, TN, TX, VA, WA, and WY. The authors assume full responsibility for analyses and interpretation of these data.

## Additional information

### Competing interests

Philip W Kantoff: reports the following disclosures for the last 24-month period: he has investment interest in Convergent Therapeutics Inc, Cogent Biosciences, Context Therapeutics LLC, DRGT, Mirati, Placon, PrognomIQ, SnyDevRx and XLink, he is a company board member for Context Therapeutics LLC and Convergent Therapeutics, he is a company founder for XLink and Convergent Therapeutics, and is/was a consultant/scientific advisory board member for Anji, Candel, DRGT, Immunis, AI (previously OncoCellMDX), Janssen, Progenity, PrognomIQ, Seer Biosciences, SynDevRX, Tarveda Therapeutics, and Veru, and serves on data safety monitoring boards for Genentech/Roche and Merck. He reports spousal association with Bayer. Giovanni Parmigiani: reports the following disclosures for the last 24-month period: he had investment interest in CRA Health; he is a co-founder and company board member of Phaeno Biotechnology; he is a consultant / scientific advisory board member for Konica-Minolta, Delfi Diagnostics and Foundation Medicine; he serves on a data safety monitoring board for Geisinger. None of these activities are related to the content of this article. The other authors declare that no competing interests exist.

### Funding

| Funder | Grant reference number | Author |
|---|---|---|
| National Cancer Institute | U01 CA167552 | Lorelei A Mucci |
| National Cancer Institute | P50 CA090381 | Philip W Kantoff<br>Massimo Loda<br>Lorelei A Mucci |
| National Cancer Institute | P50 CA211024 | Massimo Loda |
| National Cancer Institute | P30 CA008748 | Konrad H Stopsack<br>Philip W Kantoff |
| National Cancer Institute | P30 CA006516 | Massimo Loda<br>Lorelei A Mucci |
| National Cancer Institute | 5R37 CA227190-02 | Svitlana Tyekucheva<br>Giovanni Parmigiani<br>Lorelei A Mucci<br>Kathryn L Penney |
| National Cancer Institute | R03 CA212799 | Molin Wang |
| National Cancer Institute | R35 CA212799 | Molin Wang |
| National Cancer Institute | R01 CA131945 | Massimo Loda |

| Funder | Grant reference number | Author |
|---|---|---|
| DOD Prostate Cancer Research Program | W81XWH-18-1-0330 | Konrad H Stopsack |
| Prostate Cancer Foundation | Young Investigator Award | Konrad H Stopsack<br>Stephen P Finn<br>Tamara L Lotan<br>Lorelei A Mucci<br>Kathryn L Penney |

The funders had no role in study design, data collection and interpretation, or the decision to submit the work for publication.

## Author contributions
Konrad H Stopsack, Conceptualization, Data curation, Formal analysis, Funding acquisition, Methodology, Software, Visualization, Writing – original draft; Svitlana Tyekucheva, Molin Wang, Investigation, Methodology, Writing – review and editing; Travis A Gerke, Investigation, Resources, Software, Validation, Writing – review and editing; J Bailey Vaselkiv, Data curation, Investigation, Validation, Writing – review and editing; Kathryn L Penney, Tamara L Lotan, Investigation, Writing – review and editing; Philip W Kantoff, Investigation, Supervision, Writing – review and editing; Stephen P Finn, Michelangelo Fiorentino, Massimo Loda, Investigation, Resources, Writing – review and editing; Giovanni Parmigiani, Lorelei A Mucci, Funding acquisition, Investigation, Methodology, Supervision, Writing – review and editing

## Author ORCIDs
Konrad H Stopsack ![ORCID] http://orcid.org/0000-0002-0722-1311
J Bailey Vaselkiv ![ORCID] http://orcid.org/0000-0001-7702-9504

## Ethics
Written informed consent was obtained from all participants, and the study protocol was approved by the institutional review boards of the Brigham and Women's Hospital and Harvard T.H. Chan School of Public Health (IRB 19-1430), and those of participating registries as required.

## Decision letter and Author response
Decision letter https://doi.org/10.7554/eLife.71265.sa1
Author response https://doi.org/10.7554/eLife.71265.sa2

# Additional files

## Supplementary files
• Supplementary file 1. Tables, Figures, and Source Code. (**a**) Interaction terms to test for multiplicative effect modification, that is whether batch effects more strongly affect tumors with more extreme clinical/pathological characteristics. The table shows point estimates (differences in biomarker levels), 95% confidence interval bounds, p-values, and false-discovery rates (FDR, in ascending order) for interaction terms between the within-batch normalized biomarker level and the potential effect modifier in linear models that have absolute biomarker levels in standard deviation units per biomarker as the outcome and also include main effects for the biomarker and the effect modifier (terms not shown). (**b**) ICC for between-batch variance for uncorrected biomarker levels ("1 Raw") and biomarker levels after applying different correction methods. (**c**) Results from plasmode simulation according to type of induced batch effect, using the data correlation structure "moderate confounding." For three fixed ("true") hazard ratios for the biomarker–outcome association (1/3, 1, and 3), the observed hazard ratios after batch correction (with 95% confidence intervals) are shown. (**d**) Results from plasmode simulation according to data correlation structure, using the batch effect "mean and variance." For three fixed ("true") hazard ratios for the biomarker–outcome association (1/3, 1, and 3), the observed hazard ratios after batch correction (with 95% confidence intervals) are shown. (**e**) Gleason grade—biomarker associations according to batch effect correction method. Point estimates from unadjusted linear regression models for biomarker values with Gleason score categories per 1 "grade group" increase as the predictor are shown (with 95% confidence intervals). For $log_e$-transformed markers like Ki-67, estimates are differences on the $log_e$ scale. (**f**) Biomarker levels and lethal disease according to batch effect correction method. Hazard ratios (with 95%

confidence intervals) per one standard deviation increase in the biomarker (linear) from unadjusted Cox regression models are shown. (**g**) Biomarker levels and lethal disease according to batch effect correction method. Unlike in the preceding table, the hazard ratios (with 95% confidence intervals) are contrasts comparing extreme quartiles (fourth compared to first quartile) from unadjusted Cox regression models.

• Transparent reporting form

• Source code 1. Analytical code and output in R Markdown format that produced all figures, figure supplements, tables, and data mentioned in the text.

### Data availability

The batchtma R package is available at https://stopsack.github.io/batchtma and the Comprehensive R Archive Network (CRAN). Code used to produce results this manuscript is at https://github.com/stopsack/batchtma_manuscript (copy archived at swh:1:rev:a588f10906f8685b055e5a6f0a487f5f850d13bc). Data are available for analysis on the Harvard FAS computing cluster through a project proposal for the Health Professionals Follow-up Study (https://sites.sph.harvard.edu/hpfs/for-collaborators).

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
