## [Editor Report]

Tissue microarrays (TMA) have become a mainstay in clinical and basic research, for both discovery and validation of biomarkers. This manuscript provides relevant methodological considerations for cancer researchers investigating tissue-biomarkers using TMAs. A comprehensive investigation was conducted using a combination of analytic approaches using empirical data and simulated data to support key findings and conclusions. The authors approach the possible sampling variation in a thoughtful way, not only quantifying the issue systematically, but working towards a solution.

---

## [Decision Letter]

**Decision letter after peer review:**

Thank you for submitting your article "Extent, impact, and mitigation of batch effects in tumor biomarker studies using tissue microarrays" for consideration by *eLife*. Your article has been reviewed by 2 peer reviewers, and the evaluation has been overseen by Maria Spies as a Reviewing Editor and Eduardo Franco as the Senior Editor. The reviewers have opted to remain anonymous.

Essential revisions:

The reviewers and the reviewing editor agree that this is an important and well executed work. The reviewers, however have identified several technical details that need to be addressed (see the individual reviews below):

Specifically,

1) In addition to pre-analytic phase, batch effects may be introduced at analytic and post-analytic phases of investigation. The authors need to provide the reader a sense of how the analytic factors influence batch variation (see comment by reviewer 1).

2) Discuss how the TMAs construction protocols may influence the batch effects (see comment by reviewer 1).

3) Address the reviewer 2's point about avoiding overfitting.

4) Address how the tumor heterogeneity would affect the performance of batchma approach.

*Reviewer #1 (Recommendations for the authors):*

Based on the first paragraph of the methods, it seems the overall TMA set included 14 different TMA blocks labeled A through N as illustrated in Figure 1, while in Figure 3, panel B, it appears as if multiple blocks/slides are included in a single TMA (e.g., A, H, K, N). It would be useful to provide a clear description of the actual unit of analysis with respect to whether a single TMA consisted of multiple blocks, or if each TMA (e.g., A-N) was a single block. This would be particularly useful information given potential batch effects could be introduced during the analytic phase as not all 14 TMAs would be assayed simultaneously, or not at all for smaller sets that could be assayed in a single run.

As the authors only investigated whether pre-analytic factors such as tumor stage, grade, and date of diagnosis were associated with the observed batch effects it should be acknowledged that batch effects could be introduced during the pre-analytic, analytic, and post-analytic phases of biomarker investigations. For example, the authors could compare the ICCs for biomarkers assessed using software vs. eye, and % cells vs. area to provide the reader a sense of how these analytic factors influence between batch variation.

Further, it should also be acknowledged that the six statistical approaches to correcting the observed batch-effects were evaluated agnostically with respect to the source of the systematic measurement error for each biomarker. The authors may want to comment on whether these approaches, or others not evaluated, are suited for situations in which the source of the batch effect is known.

While the TMA set was nested within two large epidemiologic cohorts and constructed using standard protocols on a rolling basis as new events occurred and tissue was acquired, its not clear whether the null association between the pre-analytic factors and the batch effects observed in the 20 biomarkers could be generalizable to TMAs constructed using different protocols leveraging archived tumor samples and to other tissue biomarkers that may be influenced by the tumor microenvironment such that TMAs may not be ideal for certain biomarker investigations.

*Reviewer #2 (Recommendations for the authors):*

– Generally this reviewer does not make major comments on structure, but the Results section jumps from Figure 1 to Figure 4 to Figure 2. It makes sense in the narrative of the story, but figures should be reorganized to suite the order of discussion. It is a detriment to the work the authors have put in and shown.

---

## [Author Response]

Essential revisions:The reviewers and the reviewing editor agree that this is an important and well executed work. The reviewers, however have identified several technical details that need to be addressed (see the individual reviews below):Specifically,1) In addition to pre-analytic phase, batch effects may be introduced at analytic and post-analytic phases of investigation. The authors need to provide the reader a sense of how the analytic factors influence batch variation (see comment by reviewer 1).2) Discuss how the TMAs construction protocols may influence the batch effects (see comment by reviewer 1).3) Address the reviewer 2's point about avoiding overfitting.4) Address how the tumor heterogeneity would affect the performance of batchma approach.

Thank you for the review and the opportunity to revise the manuscript. Please note our responses to reviewer comments in detail below.

Reviewer #1 (Recommendations for the authors):Based on the first paragraph of the methods, it seems the overall TMA set included 14 different TMA blocks labeled A through N as illustrated in Figure 1, while in Figure 3, panel B, it appears as if multiple blocks/slides are included in a single TMA (e.g., A, H, K, N). It would be useful to provide a clear description of the actual unit of analysis with respect to whether a single TMA consisted of multiple blocks, or if each TMA (e.g., A-N) was a single block. This would be particularly useful information given potential batch effects could be introduced during the analytic phase as not all 14 TMAs would be assayed simultaneously, or not at all for smaller sets that could be assayed in a single run.

Each tissue microarray (TMA) was one block, named “A” through “N” in the manuscript, and was the unit of analysis throughout. We have edited the manuscript to clarify this point.

Modifications to the manuscript:

Results, first paragraph, first sentence (additions): “To evaluate the presence of batch effects in studies using TMAs, we studied tumor tissue from 1,448 men with primary prostate cancer on 14 TMAs (labeled “A” through “N”), each including multiple tumor cores from 47 to 158 patients per TMA (Figure 1).”

Results, second paragraph, second sentence (additions): “Biomarker values showed noticeable between-TMA variation, despite immunohistochemical staining having been conducted at the same time for all 14 TMAs.”

As the authors only investigated whether pre-analytic factors such as tumor stage, grade, and date of diagnosis were associated with the observed batch effects it should be acknowledged that batch effects could be introduced during the pre-analytic, analytic, and post-analytic phases of biomarker investigations. For example, the authors could compare the ICCs for biomarkers assessed using software vs. eye, and % cells vs. area to provide the reader a sense of how these analytic factors influence between batch variation.

We agree that batch effects can be introduced during any study phase, and that it is important to acknowledge this. We now do so in the discussion. In our analysis, while we focus on pre-analytic factors, we have standardized analytic and post-analytic factors to the best of our ability, leaving no known sources of variance for quantitative exploration. However, the potential exists for different approaches to image analysis to affect the extent and nature of pre-analytic batch effects. We have also added the terminology of “pre-analytic, analytic, and post-analytic” to the relevant section of the Discussion.

As a team, we discussed to what extent we could attribute higher between-TMA variance to factors such as the quantification approach (software vs. eye) or the staining measure (intensity vs. % “positive” vs. area positive). Ultimately, given the sample size of “only” 20 biomarkers (three of which were analytically identical, but had quite different ICCs; see Results, second section, third paragraph), we do not feel comfortable highlighting any contrasts, given the substantial statistical uncertainty around ICC estimates and a risk of overinterpretation. We are interested in this problem but addressing it rigorously would be beyond the scope of this paper.

Modifications to the manuscript:

Discussion, sixth paragraph (addition): “In multidisciplinary team discussions (10), it may be possible to directly pinpoint the source of batch effects and eliminate its cause. All study steps, including the pre-analytic, analytic and post-analytic phases, should be considered. If sources of batch effects can be identified, it is preferable that they be addressed directly during the pre-analytical or analytical phase, rather than applying the post-analytical methods that we have described here and that may not adequately incorporate knowledge on the source of batch effects. For example, if immunohistochemical staining was performed separately for each TMA […] at once. Imaging of pathology slides can also be a source of batch effects (11), as could be image analysis.”

To Results, fourth paragraph (previously starting with “The method of scoring…”), we added the following sentence at the beginning: “Some biomarkers were stained using automated staining systems, other stains were done manually (Figure 2). Moreover, the method of scoring…” At the end of that paragraph, we added: “Our data do not allow distinguishing which of these approaches, if any, were less prone to batch effects.”

Further, it should also be acknowledged that the six statistical approaches to correcting the observed batch-effects were evaluated agnostically with respect to the source of the systematic measurement error for each biomarker. The authors may want to comment on whether these approaches, or others not evaluated, are suited for situations in which the source of the batch effect is known.

We prefer directly addressing the source, if known; how to do that and whether the methods we describe would be applicable is probably best decided on a case-by-case basis. We now alert the reader to the fact that none of mitigation approaches leverages information about the source.

Modifications to the manuscript:

Results, third section (“Mitigation of batch effects”), first paragraph (addition): “Two mitigation approaches, batch means (approach 2) and quantile normalization (approach 6), assumed no true difference between TMAs is arising from patient and tumor characteristics, while all other approaches attempted to retain such differences between TMAs. It is possible that the choice of mitigation approaches may be optimized using knowledge of the source of the batch effect. This would be the case if each method “specialized” in mitigating effect from specific sources. We have not investigated this possibility here. Overall, correlations between values …”

See the previous point for modifications to paragraph six of the Discussion.

While the TMA set was nested within two large epidemiologic cohorts and constructed using standard protocols on a rolling basis as new events occurred and tissue was acquired, its not clear whether the null association between the pre-analytic factors and the batch effects observed in the 20 biomarkers could be generalizable to TMAs constructed using different protocols leveraging archived tumor samples and to other tissue biomarkers that may be influenced by the tumor microenvironment such that TMAs may not be ideal for certain biomarker investigations.

We agree.

Modifications to the manuscript: Discussion, sixth paragraph (addition): “In our study, both approaches indicated that the largest share of between-TMA differences was likely due to batch effects rather than true differences between tumors on different TMAs. However, one should not simply assume this to be the case in other settings, and also explore between-tumor differences as one source of between-TMA differences.”Reviewer #2 (Recommendations for the authors):– Generally this reviewer does not make major comments on structure, but the Results section jumps from Figure 1 to Figure 4 to Figure 2. It makes sense in the narrative of the story, but figures should be reorganized to suite the order of discussion. It is a detriment to the work the authors have put in and shown.

We have reordered the supplementary figures to link to a main figure, and all figures are now numbered in the order they are first mentioned.